# Machine Learning-Assisted Identification of Single-Layer Graphene via Color Variation Analysis

**DOI:** 10.3390/nano14020183

**Published:** 2024-01-12

**Authors:** Eunseo Yang, Miri Seo, Hanee Rhee, Yugyeong Je, Hyunjeong Jeong, Sang Wook Lee

**Affiliations:** 1Department of Physics, Ewha Womans University, Seoul 03760, Republic of Korea; eunseo1092@ewhain.net (E.Y.); smr871106@gmail.com (M.S.); haneerhee@naver.com (H.R.); yugyeongje@gmail.com (Y.J.); hjjeongssi@gmail.com (H.J.); 2Department of Artificial Intelligence and Software, Ewha Womans University, Seoul 03760, Republic of Korea; 3Department of Medicine, Kyung Hee University College of Medicine, Seoul 02447, Republic of Korea; 4Department of Mathematical Sciences, Seoul National University, Seoul 08826, Republic of Korea

**Keywords:** machine learning, graphene, support vector machine

## Abstract

Techniques such as using an optical microscope and Raman spectroscopy are common methods for detecting single-layer graphene. Instead of relying on these laborious and expensive methods, we suggest a novel approach inspired by skilled human researchers who can detect single-layer graphene by simply observing color differences between graphene flakes and the background substrate in optical microscope images. This approach implemented the human cognitive process by emulating it through our data extraction process and machine learning algorithm. We obtained approximately 300,000 pixel-level color difference data from 140 graphene flakes from 45 optical microscope images. We utilized the average and standard deviation of the color difference data for each flake for machine learning. As a result, we achieved F1-Scores of over 0.90 and 0.92 in identifying 60 and 50 flakes from green and pink substrate images, respectively. Our machine learning-assisted computing system offers a cost-effective and universal solution for detecting the number of graphene layers in diverse experimental environments, saving both time and resources. We anticipate that this approach can be extended to classify the properties of other 2D materials.

## 1. Introduction

Graphene is a promising 2D material that has been studied intensively due to its interesting physical and material properties [1,2,3,4]. Graphene flakes can be detected by observing images of optical microscopes ever since the discovery of their ability to be separated by Scotch tape exfoliation [5]. The flakes are randomly placed through Scotch tape exfoliation on SiO_2_ with a certain thickness, and the image of those flakes is captured by an optical microscope. Based on the Fresnel law, a specific thickness of the SiO_2_ makes a maximum contrast so that a single layer of graphene can be visible even though there is only a subtle color difference between the graphene and the substrate. In this case, SiO_2_ with thicknesses of 300 nm and 100 nm, known to be suitable for the visual detection of graphene, are usually used [6]. There are several techniques for estimating how many layers the graphene flake has after it is exfoliated or synthesized on the substrate [7]. Among these methods, Raman spectroscopy and atomic force microscopy (AFM) are the most commonly applied methods to identify the number of graphene layers [8,9,10,11,12]. However, the above-mentioned techniques not only require specialized and expensive equipment but also take a considerable amount of time for measuring and analyzing to determine the number of graphene layers. The thickness of graphene from the AFM measurement might be exaggerated due to the interaction between the AFM tip and the substrate, especially for single-layer graphene. In any cases of identifying the number of layers or finding single-layer graphene, the first step of work should be to confirm the location of the graphene fragment through an optical microscope observation. At this point, by observing the color of the graphene fragment on the substrate, one can estimate the thickness of the graphene to some extent. Highly trained researchers can even differentiate subtle color differences between the substrate and the graphene fragment and can identify single-layer graphene through optical microscope observation itself. In other words, the observation of graphene flakes comparing the color differences between the flake and its surroundings with a “well-trained” human eye might be a more accurate, faster, and more cost-effective method of identifying graphene layers.

Thanks to many open source software tools, it has become easier to obtain numerical information about graphene flakes (such as material location, shape, color differences from the surroundings, etc.) from image data. Furthermore, recent advancements in computer resources have provided greater storage capacity and faster data processing capabilities. This enables the computer system to learn from input data, analyze patterns, and subsequently make decisions, classifications, and predictions tailored to specific tasks by applying machine learning algorithms. Machine learning has led to huge achievements in various fields like natural language processing, speech recognition, and in particular, image recognition [13,14,15,16,17,18,19]. At the same time, there have been several successful studies reported to detect graphene automatically by utilizing various machine learning algorithms on optical image datasets [20,21,22,23,24,25]. However, there have been some challenges with diverse experimental conditions, such as the types of optical microscopes, lighting conditions, magnification levels, and background colors due to the different thicknesses of the substrate. All of these factors have caused inconsistent performance. Thus, relying solely on traditional machine learning and deep learning algorithms may not consistently produce excellent performance in the task of classifying graphene. In addition, some algorithms require large datasets and manual labeling efforts for training deep learning models. Consequently, there is a need for new ideas that are easy to use and guarantee consistent performance under any experimental setup conditions.

A well-trained researcher, on the other hand, can adapt to the various experimental environments described above, learn according to the situation, and distinguish the color differences between the substrate and the graphene flake, thereby detecting graphene. Therefore, it is expected that by employing data processing and machine learning algorithms which mimic the methods of human researchers who work on finding graphene through an optical microscope, it will be possible to classify single-layer graphene from graphene flake images obtained in various experimental environments.

In this study, based on the idea that skilled human operators can identify graphene solely through optical images without additional methods, such as Raman spectroscopy, in diverse experimental setup conditions, we devised a machine learning-assisted computing system that mimics this method. We extracted color differences between the flake and the surrounding substrate from flakes in the image data, calculating the average and standard deviation of color differences per flake, which we then utilized in machine learning algorithms. Consequently, we discovered that we could achieve an F1-Score of over 0.90 by simulating the human color discrimination method. It can be universally used in various experimental settings for automatically detecting graphene, and further, we believe this method could apply to other 2D materials.

## 2. Modeling and Methods

### 2.1. Our Approach to Identifying Single-Layer Graphene

Figure 1 shows the overall procedure for identifying a single graphene layer by both a human researcher at the laboratory (upper grey box) and by our computer process and machine learning algorithm, mimicking how the human behaves (lower light blue box). To distinguish the number of graphene layers in the laboratory, human researchers perform their experimental work as follows. First, the position of the graphene is identified through optical microscope observation. Next, the outline of the graphene is recognized from the image. Then, the number of graphene layers is estimated by comparing the color differences between the edges of the graphene and its surroundings. Experienced researchers, through extensive training, are able to empirically determine whether the observed graphene is single-layer, double-layer, or multilayer, based on the color differences between the substrate and the graphene.

We then adopted the above-mentioned human being’s working process to our algorithm to identify a single layer of graphene from the optical microscope images of graphene flakes. To identify objects in the image, first, we employed a data acquisition and comparison process to detect data points for calculating color differences according to the characteristics of each step depicted in Figure 1. After an optical image is inputted, our system acquires vertices to recognize the outlines of flakes in the image (labeling process). Then, these vertices are connected to form the outlines, and the RGB values of pixels at positions a certain distance away from the outlines are extracted (recognition of outline) [26]. The color difference between the outlines and the surrounding substrate is calculated (data acquisition and calculation process). Based on the RGB difference information obtained from the data processes, the number of graphene layers is classified through a trained machine learning model (decision-making). In this case, we utilized a support vector machine (SVM), one of the well-established machine learning models, to solve classification and regression tasks to determine the number of graphene layers [27]. We optimized the SVM model by training it with our optical microscope image dataset. We achieved an F1-Score of over 0.90 in binary classification, effectively categorizing the training dataset into a single-layer class and a multilayer class.

### 2.2. Methods

#### 2.2.1. Generating Graphene Image Datasets and Data Labeling

We used 45 optical microscope images of graphene flakes placed on SiO_2_. The color of the substrate varies depending on the optical microscope’s magnification and illuminating light source. In our image data, we observed roughly two main color tones: pink and green. We differentiated these two background colors and separately applied them to the machine learning algorithm. To achieve robust classification performance, we collected graphene images captured by multiple human researchers in diverse experimental environments, such as those containing different background colors or with various magnifications. All the images were directly imported into our procedure without resizing, resolution enhancement, color normalization, and so on. The method we propose can accept most microscope images in various conditions, enabling a universal solution for differentiating single-layer graphene out of deposited flakes.

Besides the above-mentioned aspects in the experimental environment of image taking, the optical microscope images of graphene can be sensitive to the type and brightness of the microscope’s light source. Additionally, if the substrate is not perfectly aligned perpendicularly to the light source and has a slight tilt in the vertical direction, the colors at one end of the substrate and those the other end may vary slightly. In the case of a single-layer graphene image, in particular, the color of graphene can be sensitively varied according to the angle of the light source and the substrate. Therefore, traditional edge detection algorithms such as traditional Canny edge detection methods [28] in computer vision and image processing tend to fail to accurately capture the boundaries, since there is very little color difference between single-layer graphene and the surrounding substrate. To identify the boundaries of flakes in the collected images, as shown in Figure 2a, we performed labeling that provided answers to input data using the VGG software developed by the Visual Geometry Group [29] for 45 images. This tool assists in adding annotations and assigning labels to image and video data, playing a crucial role in the data preparation phase for training and evaluating computer vision models. Based on the layer information of each flake, confirmed through Raman spectroscopy, we assigned labels to determine whether each flake is single-layer or multilayer. The labeled data were stored in JSON file format [30].

#### 2.2.2. Recognition of Graphene Outline and Its Surroundings

To mimic the process of human operators identifying the boundaries of the graphene flakes and examining the color differences around the boundaries, we implemented a data extraction code that automatically identifies the boundaries of flakes based on the provided labels. The system collected the pixel positions (x, y) of the flake’s boundary in the image, referring to the JSON file containing the labeling information. After that, P_in_ and P_out_ points were defined, as shown in Figure 2b, at the outside (x_out_, y_out_) and inside (x_in_, y_in_) of the graphene flake with certain distances from its boundary. We set 14-pixel points as an optimal distance value from the boundary of the flake to define the pair of inside/outside positions to calculate color differences. It was found that if the distance is less than the chosen pixel distance, the RGB value could be distorted due to the interference of microscope images close to the edge of graphene flakes. Also, if the distances from the edge of the graphene are great, the RGB value differences between P_in_ and P_out_ could be larger due to the brightness differences that occur with the image under the test. In our data acquisition process, the P_in_ and P_out_ pairs obtained from the narrow area of the graphene flake, where widths are less than 14 pixels, were ruled out.

The position information for the selected pairs was then uploaded to our database. Figure 2b shows the process of collecting the positions of training data used in our algorithm. The same procedure was applied to every outline pixel of each flake, resulting in the collection of hundreds of pixel coordinates per graphene flake.

#### 2.2.3. Data Acquisition

After defining the pixel positions around the graphene boundaries, the RGB values were extracted from each pair of pixels. Then, these color information data were uploaded to the database. Inspired by the ability of experienced human operators to distinguish layer numbers based on color differences, we designed the system to automatically calculate the color difference between the surrounding substrate and the flake boundaries. Figure 3 shows three-dimensional RGB value distribution mappings obtained from the stored pixels and the optical microscope images, with the defined pixel positions corresponding with the 3D RGB coordinate system. In Figure 3, the P^1^_in/out_ and P^4^_in/out_ values, as indicated, are obtained in the vicinity of flakes considered to be single-layer graphene on a pink background and a green background, respectively. The P^2^_in/out_ and P^3^_in/out_ values, on the other hand, represent the values obtained in the vicinity of multilayer graphene. The RGB values on the pink-background substrate are represented as yellow dots, while the values within single/multilayer graphene flakes are indicated as pink/red dots, respectively. Similarly, for the green-background substrate, the RGB values on the substrate are represented as light green, and the values within single/multilayer graphene flakes are shown as green/blue.

As mentioned earlier, the distribution of RGB data can vary depending on experimental conditions, such as the angle of the light source and variations in substrate color. As seen in Figure 4, it is evident that due to variations in background colors and resulting brightness differences, even the same single-layer graphene may not be easily distinguishable in RGB coordinates. Furthermore, within the same image, variations in lighting can lead to differences in brightness, resulting in a noticeable color value shift in the RGB coordinate for inner and outer points along the outline of the graphene flake. Therefore, when employing a machine learning classification algorithm trained on the RGB values of both the object and the substrate, it becomes exclusively applicable under the condition of identical background colors. However, the differences in RGB values between the substrate and the single-layer graphene exhibit a similar pattern. Therefore, instead of classifying the RGB space using machine learning, we accumulated RGB difference values (ΔRGB) and used them for training the machine learning model to determine the layer numbers by learning the distances between them. Subsequently, the RGB value differences were calculated and uploaded to the database. As can be seen in Figure 3, the lines connecting points represent the RGB differences, and single-layer graphene and multilayer graphene can be clearly separated in the RGB color space. This method is similar to how a trained human operator works in the lab to determine single-layer graphene by differentiating the color differences between the substrate and the graphene flake.

#### 2.2.4. Application of Machine Learning Model

We chose support vector machine (SVM) [31,32] to classify the input data into single-layer and multilayer classes. SVM is recognized as one of the models with excellent performance and generalization ability. It maps the given data into a higher-dimensional space to find a decision boundary with the maximum margin between classes. This model effectively separates the data and optimizes prediction performance. SVM utilizes support vectors, the closest data points to the decision boundary, to find the optimal decision boundary based on their positions and distances.

We employed the k-fold cross-validation evaluation method [33], where k represents the number of data partitions and training iterations, to assess the consistency of our model’s performance across each fold. We used k = 5, which means that the data are divided into five parts, where four parts are used as the training dataset, and the remaining part is used as the test dataset. The test and training sets are alternated based on the number of data partitions, with a 1:4 ratio. Furthermore, we used the F1-Score as an evaluation metric to precisely assess our model’s performance in classification tasks. The F1-Score captures the harmonic mean of precision and recall. Precision, indicating the ratio of true positives among the model’s positive predictions, is utilized to minimize false positives. Recall, representing the ratio of accurately predicted positives among the actual positives, is employed to minimize false negatives. A high F1-Score not only indicates accurate predictions but also signifies a model that is effective in minimizing missed positive samples and is not sensitive to data imbalance between classes.

## 3. Results and Discussion

### 3.1. Training Data Distribution and Decision Boundary

To differentiate the single-layer graphene, we incorporated the standard deviation value as a training parameter as well as the average value of RGB differences. Figure 5a shows the distribution of average color difference and standard deviation data between graphene flakes and their surrounding pink/green substrates, as well as the decision boundary trained by the SVM model. For single-layer graphene flakes on the pink-based background, the distribution showed a mean of 13.09 and a standard deviation of 6.13. For multilayer flakes, the distribution showed a mean of 37.02 and a standard deviation of 30.50. In the case of flakes with a green-based background, for single-layer flakes, the distribution showed a mean of 20.80 and a standard deviation of 10.34. For multilayer flakes, the distribution showed a mean of 107.05 and a standard deviation of 43.90.

As shown in Figure 5b, the classification between single-layer and multilayer, based solely on the average color difference, was not distinctly clear. However, when considering the standard deviation of the color difference, the classification between the two classes became notably distinct. To elaborate, in Figure 5a, on the pink-based substrate, there are a few data points where the color difference average suggests a higher likelihood of classifying them as single-layer, but they were correctly identified as multilayer due to a higher standard deviation. Upon closer examination of the data, it was observed that these flakes had complex geometry in an image, which resulted in incorrect distinctions between P_in_ and P_out_ pairs. There were some error cases where color differences were unexpectedly measured inaccurately due to flakes being located at unpredicted positions. One of the examples of this error case is shown in the inset of Figure 5a, where two bilayer graphene flakes were located close to each other. As indicated with a red dotted circle, the P_out_ position could be defined at the inside of the neighboring graphene flake. In this case, the average value of ΔRGB could be decreased due to the mis-defined pair of positions, which are not automatically ruled out. However, as the standard deviation increased due to the unexpected variations, classification remained feasible despite these errors. Therefore, we found that the standard deviation of color difference serves as a crucial parameter for distinguishing between single-layer and multilayer graphene.

### 3.2. Model Performance

The F1-Score results showed a score of 0.90 with only 60 flakes’ information in images with a pink-based background, and a score of 0.92 with only 50 flakes’ information in images with a green-based background. The ability to achieve a high score with a small number of objects in each set of 45 images is advantageous for practical applications and demonstrates the method’s efficiency.

In Figure 5a, it can be observed that graphene on green-based substrates has a wider margin and is classified more distinctly than graphene on pink-based substrates. This result is akin to that of humans naturally perceiving green tones more effectively and better discerning subtle color variations within the green spectrum. Our data also support the research finding that color differentiation is more clearly distinguished within the green spectrum [34].

Our goal was to achieve robust performance with a small amount of data for the accessibility of human researchers. Therefore, we conducted tests to determine the minimum number of labeled flakes to be included in the database for training. For both pink- and green-toned images, we experimented with varying the number of flakes, ranging from 30 to 60 (pink) and 20 to 50 (green), as shown in Figure 6. In the case of pink-based images, we observed that performance approached 0.90 when using a minimum of 60 flakes. Conversely, given their strong classification performance as indicated by the RGB distribution, it was not necessary to use as many as 60 flakes for green-based images. Even with 40 flakes, green-based images achieved a classification score of 0.90. This indicates that when classifying graphene using our proposed method, a score of over 0.90 can be maintained even with a small dataset.

## 4. Conclusions

This study proposes a solution for distinguishing the number of graphene layers, even under various experimental conditions and variations in illumination. We addressed issues such as differences in illumination, the need for collecting a large amount of data, extremely time-consuming manual labeling tasks, and errors in automatic labeling. To achieve this, we applied a data acquisition method and machine learning techniques that mimic the human method of using a microscope to determine single-layer graphene through color differences. By utilizing the color difference information around flakes, our method demonstrates robust classification capabilities even in the presence of illumination variations within images. Moreover, with a small amount of data, we were able to maintain an F1-Score of over 0.90. Additionally, since this approach is based on color difference, it has the potential to extend beyond graphene and enable the differentiation of other 2D materials as well. Recently, 2D material-based electronic devices have been practically incorporated into the integrated CMOS system [35,36,37]. To adopt 2D materials into the fabrication process of integrated circuits, a fast and cost-effective method for distinguishing a single layer of 2D materials is required. A machine learning-based identification of a single layer of graphene and 2D materials, as presented in this work, could be applicable to a processing methodology that aligns with these requirements.

## 5. Patents

A Korean patent based on this work will be filed.

## Figures and Tables

**Figure 1 nanomaterials-14-00183-f001:**
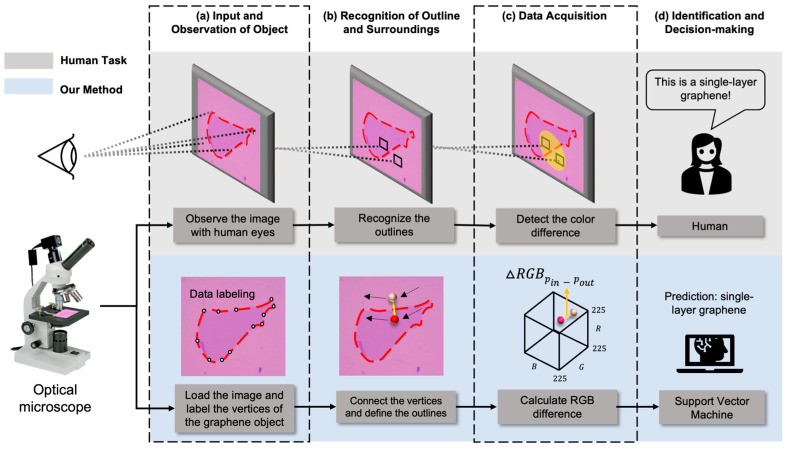
Flowchart of our algorithm mimicking the procedure in which humans distinguish the number of graphene layers: (**a**) Recognizing the positions of graphene objects within a graphene image obtained through an optical microscope (initial labeling is required for our method). (**b**) Observing the outlines of the objects. (**c**) Comparing color differences around the outlines. (**d**) Determining the number of graphene layers based on the extent of color differences and subsequently reaching a decision.

**Figure 2 nanomaterials-14-00183-f002:**
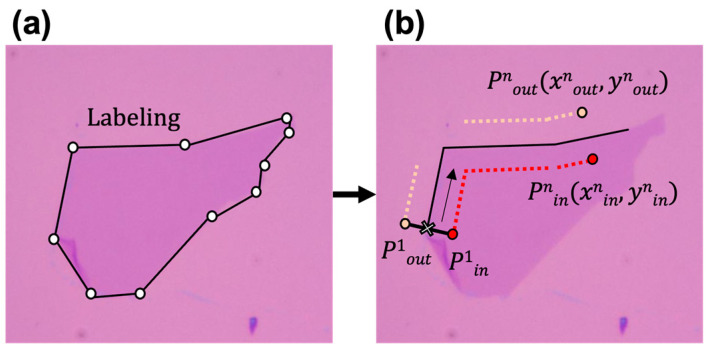
Process of collecting the positions of training data used in our algorithm. (**a**) Labeling of graphene area by saving the vertices of graphene objects using VGG Annotator. (**b**) Defining pairs of pixel positions along the labeled graphene area for comparing the color difference between the substrate (denoted by subscription “out”) and the graphene (denoted by subscription “in”).

**Figure 3 nanomaterials-14-00183-f003:**
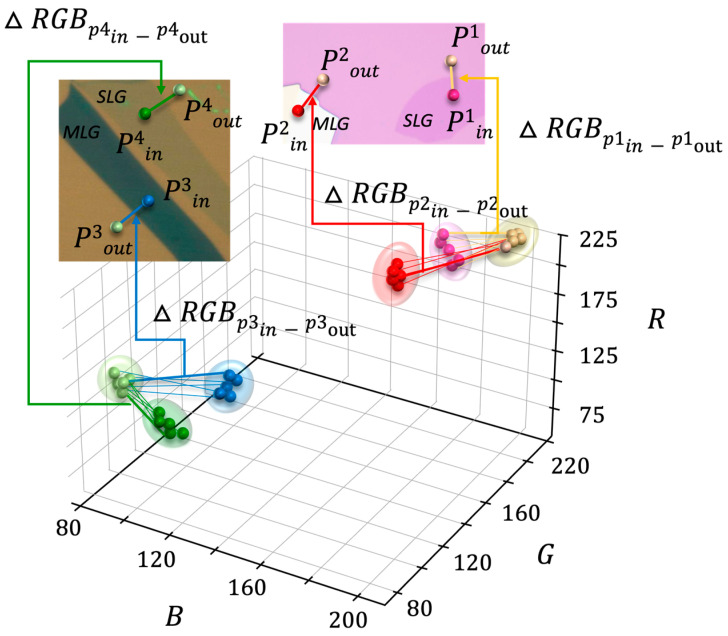
An example representation of the RGB values of a graphene object within image data in the RGB coordinate system. Optical microscope images, with the defined pixel positions, correspond with the 3D RGB coordinate system shown in the 3D plot. The line between each pair of points (P_in_, P_out_) represents the distance (ΔRGB) between the points in the RGB coordinate system. P_1_ and P_4_ depict single-layer graphene (SLG), while P_2_ and P_3_ represent multilayer graphene (MLG).

**Figure 4 nanomaterials-14-00183-f004:**
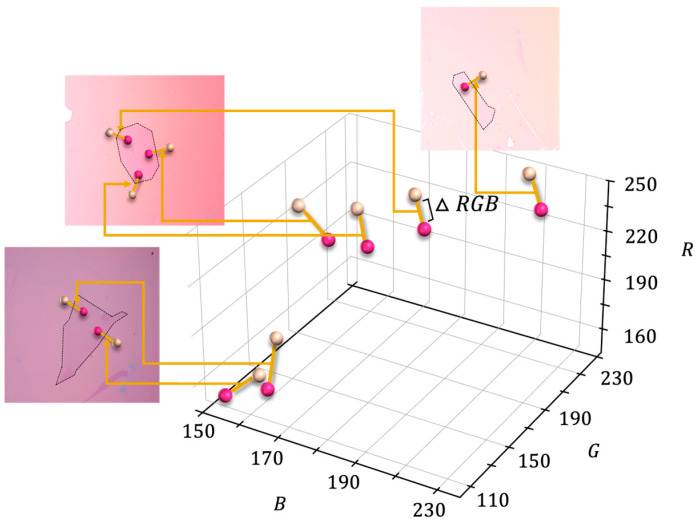
A plot of the RGB coordinates of inner and outer pixels surrounding a single-layer graphene flake in our selected optical microscope image dataset, with various pink backgrounds and different variations in lighting within an image.

**Figure 5 nanomaterials-14-00183-f005:**
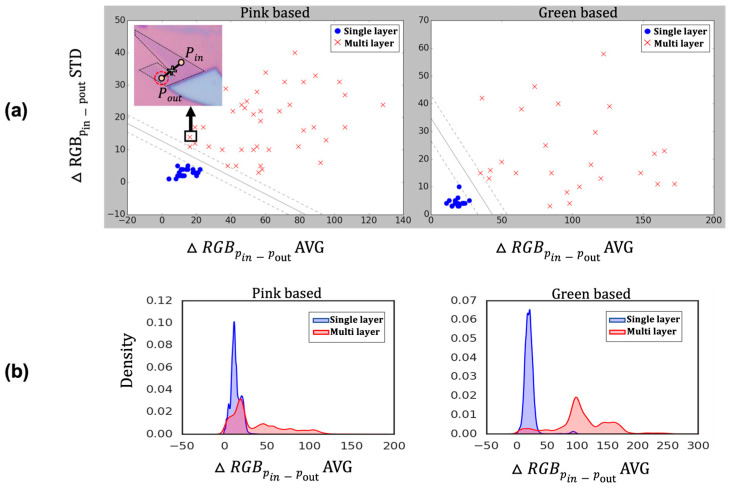
Distribution of graphene training data on the pink/green substrate and their training results. (**a**) Decision boundary results of trained SVM model classifying single-layer graphene and multilayer graphene. (**b**) Data distribution obtained through kernel density estimation. Inset of (**a**) is multilayer graphene images, of which the average ΔRGB is close to the single-layer graphene (SLG), while the standard deviation of ΔRGB is larger than the SLG.

**Figure 6 nanomaterials-14-00183-f006:**
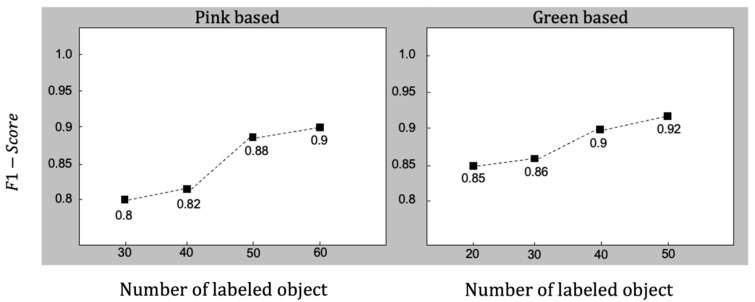
F1-Scores of the trained SVM model according to the number of labeled object samples. Each data sample represents a single-layer graphene/graphite flake used for training.

## Data Availability

The raw data of optical microscope images utilized on this study was produced by experimental works from authors. Data sets used in the current study are available from the corresopind author on reasonable request.

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
