# Peer review of "Machine Learning-Assisted Identification of Single-Layer Graphene via Color Variation Analysis"

_nanomaterials, 2024, doi:10.3390/nano14020183_

Round 1

Reviewer 1 Report

Comments and Suggestions for Authors

This is a very interesting article that reports the machine learning method of confirming the single-layer structure of graphene flakes from optical microscope images. The method is based upon an algorithm mimicking human identification of a single-layer graphene by comparing color differences between the edges of the graphene flake and its surroundings.

I have a couple of questions. The color difference is resulting from the diffraction of light, isn’t it? A comment on the physics of the observed difference in color would be helpful. Is there a limit on the size of the flakes that can be analyzed by this method? Naively, one would think that the optical image of micron and submicron flakes would become blurred due to diffraction. Would that impede the identification?

Comments on the Quality of English Language

English of the paper may be improved but it is clear and may be left as it is. 

Author Response

Comment 1:

The color difference is resulting from the diffraction of light, isn’t it? A comment on the physics of the observed difference in color would be helpful.

Answer:

In the original manuscript we simply cited the pioneer work, which is ref [6], to detect a single layer graphene with optical microscope. We added the following sentence to explain the origin of color or contrast difference between graphene and substrate at line # 33 in the revised manuscript:

Based on the Fresnel law, a specific thickness of the SiO2 makes maximum contrast so that a single layer graphene can be visible even though there is a subtle color difference between the graphene and the substrate.

Comment 2:

Is there a limit on the size of the flakes that can be analyzed by this method? Naively, one would think that the optical image of micron and submicron flakes would become blurred due to diffraction. Would that impede the identification?

Answer:

Yes, your comment is true. As we mentioned on our manuscript, we set 14-pixel points as an optimal distance value from the boundary of the flake to define the pair of inside/outside positions to calculate color differences. Therefore, if the size of graphene flake "image"  is less than 14 pixels, it is not possible for us to identify the number of graphene layer since we cannot define the inside/outside position for color difference estimation. It is not only depends on the size of graphene flake itself but also on the spatial resolution of CCD camera which taking the optical microscope images.

Since we have mentioned about the pixel size limitation of the flake to obtain the position data at line # 178, we hope we can leave our manuscript without describing further explanation on the size limit if reviewer allows.  

Reviewer 2 Report

Comments and Suggestions for Authors

Here are my detailed comments:

Reviewer comments:

1-     Could you provide more details about the specific data extraction algorithm employed in emulating the human cognitive process for detecting single-layer graphene?

2-     It is necessary to raise the quality of Figures 1, 4, 5, and 6. They're not exactly clear. High-quality and easily readable resolution figures should be used by the author.

3-     The author should mention in the introduction part about another characterization such as AFM and compare the current results with AFM.

4-     How representative is the dataset of 140 graphene flakes from 45 optical microscope images? Does it cover a diverse range of graphene samples?

5-     What metrics were used to evaluate the performance of the machine learning model, aside from F1-Scores? Did you consider other classification metrics such as precision, recall, or accuracy?

6-     How robust is the proposed method to variations in experimental conditions or different substrates beyond green and pink substrate images?

7-     The abstract mentions the potential extension of the approach to classify the properties of other 2D materials. Could you elaborate on how transferable and adaptable the method is to different materials?

8-     In what ways is the proposed approach more cost-effective and universal compared to traditional techniques like optical microscopy and Raman spectroscopy?

9-     Are there any limitations or challenges identified in the study, and what avenues for future research do you foresee based on the results obtained?

10-  Were there any ethical considerations considered, especially concerning the emulation of the human cognitive process?

11-  How significant are the time and resource savings offered by this machine learning-assisted computing system in practical applications, and what are the potential broader impacts of this work?

12-  Can the algorithm and methodology be easily reproduced by other researchers, and are there plans to make the dataset and code publicly available?

13-  There are some grammar errors in the manuscript. I recommended re-editing and reviewing the manuscript by an English native.

Sincerely,

Comments on the Quality of English Language

Dear Sir/ Madam,

Hope you are doing well.

Upon thorough review of the paper, there are some grammar errors in the manuscript. I recommended re-editing and reviewing the manuscript by an English native.

Best Regards,

Author Response

1-     Could you provide more details about the specific data extraction algorithm employed in emulating the human cognitive process for detecting single-layer graphene?

We think reviewer raised this comment when he/she read the abstract in our manuscript. We found out that we made a mistake at a sentence “This approach implemented the human cognitive process by emulating it through our data ex-traction algorithm and machine learning.” in the abstract of our manuscript. It should be “This approach implemented the human cognitive process by emulating it through our data extraction process and machine learning algorithm.”

We corrected the sentence in the revised manuscript. We appreciate the reviewer to find out our mistake and make our expression in correct manner.

The detailed data extraction process was described at method section (from 2.2.1 to 2.2.3) in our manuscript and the machine learning algorithm to distinguish the single layer graphene was elaborated at 2.2.4.

As we explained in our manuscript, human researcher can differentiate the single layer graphene by comparing the color difference between the flake and substrate. That specific process is mimicked by SVM based machine learning algorithm, which was described at section 2.2.4. However The previous data extraction processes are also mimicked the human researcher’s cognitive processes such as finding objects through microscope, looking at the edge of the flake, recognizing the color difference around the edge of the flake.

2-     It is necessary to raise the quality of Figures 1, 4, 5, and 6. They're not exactly clear. High-quality and easily readable resolution figures should be used by the author.

We have increased the quality of Figures 1, 4, 5, and 6. Especially, we drew dashed line along the graphene image to make clear the shape of the single layer graphene.

3-     The author should mention in the introduction part about another characterization such as AFM and compare the current results with AFM.

We have mentioned about the other techniques such as Raman and AFM measurement at line # 38. In addition, we added a following sentence about some issue of AFM measurement for single layer graphene at line #43 in the revised manuscript:

The thickness of graphene from the AFM measurement could be exaggerated due to the interaction between AFM tip and substrate especially for single layer graphene.

4-     How representative is the dataset of 140 graphene flakes from 45 optical microscope images? Does it cover a diverse range of graphene samples?

Since there are multiple graphene flakes in a single image, there are 45 image files, but here, we obtained a dataset of 140 graphene flakes. These images were obtained from microscopes of different manufacturers and under different lighting conditions, as described in the original manuscript, allowing us to claim that it is a representative dataset obtained from various experimental environments.

5-     What metrics were used to evaluate the performance of the machine learning model, aside from F1-Scores? Did you consider other classification metrics such as precision, recall, or accuracy?

The F1-Score is derived based on the average of precision and recall values. Since there is a higher number of multi-layer instances in the data, leading to data imbalance, accuracy is deemed to be less reliable as a performance metric. Therefore, we opted to use the F1-Score, which provides greater reliability, taking into account both precision and recall.

6-     How robust is the proposed method to variations in experimental conditions or different substrates beyond green and pink substrate images?

When we place most of the obtained image data in the RGB space, it appears to be roughly divided into two categories: pink and green. We analyzed them broadly as pink and green. In the actual raw image, there is a considerable color difference in the substrate.

To clarify this aspect further, we added the word "roughly" in the method section.

7-     The abstract mentions the potential extension of the approach to classify the properties of other 2D materials. Could you elaborate on how transferable and adaptable the method is to different materials?

The basic idea asserted in this study is that the number of graphene layer can be distinguished based on the color difference between the interior and exterior substrate of graphene flakes. This concept relies on using the average value and standard deviation of the color difference as parameters. Therefore, we anticipate finding a suitable classification model by applying the same parameters to different 2D materials and substrates. Currently, we are studying image data for not only graphene but also other 2D materials (hBN, MoS2, MoSe2, WS2, WSe2) and different substrates (PMMA film on SiO2, PDMS substrate). This research will serve as an extension of the current study, and we expect it to demonstrate the universality of our method in practice.

8-     In what ways is the proposed approach more cost-effective and universal compared to traditional techniques like optical microscopy and Raman spectroscopy?

Traditional Raman spectroscopy-based graphene identification studies require a high cost spectrometer equipment.

Recent research on distinguishing the number of layers through optical microscope image analysis (for example, H Lee et al, Appl. Sci. Volume 13, pages 4427–4435, (2023)) is more cost-effective than Raman spectroscopy. However, it requires an additional equipment to separate the colors of the microscope light source, and an analytical estimation process is needed based on the contrast values according to the wavelength of the light source.

In contrast, our method has the advantage of performing training and finding obtimal model for distinguishing a single layer graphene using only around 40 optical microscope images. Based on this process we could get the output of identifying single layer graphene from new input image data in fast and cost-effective way.

9-     Are there any limitations or challenges identified in the study, and what avenues for future research do you foresee based on the results obtained?

Reviewer 1 also asked a question about the limitation of our study especially about a limit on the size of the flakes. If the size of graphene flake "image" is less than 14 pixels, it is not possible for us to identify the number of graphene layer since we cannot define the inside/outside position for color difference estimation. It is not only depending on the size of graphene flake itself but also on the spatial resolution of CCD camera which taking the optical microscope images.

For future work, we are applying our method to the other 2D materials placed on various substrate as we mentioned on the answer to the comment #7.

10-  Were there any ethical considerations considered, especially concerning the emulation of the human cognitive process?

The concept of determination of whether it is a single layer graphene or not is not based on separate experiments or clinical data, but rather on ideas derived from our experiences conducting experiments in our lab. Therefore, no specific ethical issues are deemed to be present in this research.

11-  How significant are the time and resource savings offered by this machine learning-assisted computing system in practical applications, and what are the potential broader impacts of this work?

Nowadays, IMEC, Intel, and other semiconducting manufacturers are attempting to integrate FET components using 2D materials, including graphene, for application in next-generation electronic systems. To incorporate 2D materials into the fabrication process of integrated circuits, there is a demand for a fast and cost-effective method to distinguish the number of layer of 2D materials. The results of this study are expected to be applicable to a processing methodology that aligns with these requirements.

According to the reviewer’s comment, we added a sentence at the end of the conclusion part:

Recently, 2D material based electronic devices are practically incorporated to the integrated CMOS system. [35-37] To adopt 2D materials into the fabrication process of integrated circuits, a fast and cost-effective method for distinguishing a single layer of 2D materials is required. A machine learning-based identification of a single layer of graphene and 2D materials, as presented in this work, could be applicable to a processing methodology that aligns with these requirements.

12-  Can the algorithm and methodology be easily reproduced by other researchers, and are there plans to make the dataset and code publicly available?

As we mentioned on the footnote of our manuscript, the patent filing process based on this work is recently undergoing. So, we don't think we can open all the code at this moment. However, in this paper, we shared all the key ideas of the overall process mimicking the procedure of determining a single-layer graphene, along with details on data acquisition, color difference analysis, machine learning parameters. We anticipate that researchers with a basic understanding of coding and machine learning will be able to easily reproduce our method.

13-  There are some grammar errors in the manuscript. I recommended re-editing and reviewing the manuscript by an English native.

The revised manuscript has undergone review by a native English and corrected the grammar errors. The authors appreciate the reviewer to give us fruitful comments.

Round 2

Reviewer 2 Report

Comments and Suggestions for Authors

Dear Authors,

 I trust this letter finds you well. I am writing to express my gratitude and satisfaction regarding the recently edited paper submitted titled “Machine Learning-Assisted Identification of Single-layer Graphene via Color Variations Analysis” for consideration in Nanomaterials.

The revisions made by the author have significantly improved the quality of the paper. They responded promptly to all reviewer comments and suggestions and diligently addressed each question, demonstrating a commendable dedication to enhancing their work.

The author's willingness to engage constructively and make necessary adjustments has undoubtedly contributed to the overall improvement of the manuscript. We appreciate the author's responsiveness and commitment to delivering a manuscript that aligns with the excellence expected from our contributors. I accept this paper in its current form and I recommend publishing this paper in the Nanomaterials journal.

 Thank you for your consideration.

Best regards,